# Dermoscopy of Small Diameter Melanomas with the Diagnostic Feasibility of Selected Algorithms—A Clinical Retrospective Multicenter Study

**DOI:** 10.3390/cancers13236095

**Published:** 2021-12-03

**Authors:** Monika Slowinska, Grazyna Kaminska-Winciorek, Elzbieta Kowalska-Oledzka, Iwona Czarnecka, Robert Czarnecki, Anna Nasierowska-Guttmejer, Elwira Paluchowska, Witold Owczarek

**Affiliations:** 1Department of Dermatology, Military Institute of Medicine, Szaserow 128, 04-141 Warsaw, Poland; mslowinska@wim.mil.pl (M.S.); iczarnecka@wim.mil.pl (I.C.); epaluchowska@wim.mil.pl (E.P.); wowczarek@wim.mil.pl (W.O.); 2Evimed Medical Center, Private Dermatologic Practice, JP Woronicza 16, 02-625 Warsaw, Poland; e.kowalskaoledzka@gmail.com; 3The Skin Cancer and Melanoma Team, The Department of Bone Marrow Transplantation and Oncohematology, Maria Sklodowska-Curie National Research Institute of Oncology (MSCNRIO), Wybrzeze Armii Krajowej 15, 44-102 Gliwice, Poland; 4All4Skin, Center of Diagnostics and Treatment of Skin Diseases, Armii Krajowej 194/1, 40-750 Katowice, Poland; 5Department of Cardiology, LUX MED Oncology Limited Liability Company, Fieldorfa 40, 04-125 Warsaw, Poland; robert.czarnecki@luxmed.pl; 6Department of Pathomorphology, Central Clinical Hospital of Ministry of Internal Affairs and Administrative, Woloska 137, 02-507 Warsaw, Poland; anna.nasierowska@cskmswia.pl; 7Faculty of Medicine, Lazarski University, Swieradowska 43, 02-662 Warsaw, Poland

**Keywords:** secondary melanoma prevention, small-sized melanoma, micromelanoma, dermoscopy, diagnostic algorithm, Triage Amalgamated Dermoscopic Algorithm, 7-point checklist of dermoscopy

## Abstract

**Simple Summary:**

The early detection of melanoma determines the recovery of the patient. Dermoscopy, which is one of the diagnostic tools for pigmented lesions, is characterized by high sensitivity and specificity, giving the clinician the possibility to detect the presence of abnormal structures before their clinical presentation. There are a small number of dermoscopic analyses of pigmented lesions of less than 6 mm in diameter in the published literature. The authors attempted to identify characteristic dermoscopic structures typical for melanomas of less than 5 mm in diameter in comparison with a group of melanomas exceeding this dimension at an identical clinical stage. It was found that dermoscopy in the secondary prevention of micromelanomas (appearing mainly as brown lesions) revealed the presence of dotted or polymorphous vessels, with architectural disorder in half of cases. Moreover, spitzoid, multicomponent asymmetric or nonmelanoma-specific patterns prevailed. Knowledge of these dermoscopic features brings the clinician closer to an early diagnosis of melanoma with a diameter of 5 mm or less.

**Abstract:**

Objective: The aim of the study was to verify two hypotheses. The first concerned the possibility of diagnostic dermoscopic differentiation between cutaneous melanomas of the histopathological category in situ (pTis) and thin melanomas (pT1a) in terms of their diameter. The second assessed the diagnostic feasibility of two dermoscopic algorithms aiming to detect ≤ 5.0 mm-sized melanomas histopathologically confirmed as pTis and pT1a. Methods: Dermoscopic images of consecutive cases of histopathologically confirmed melanomas were evaluated by three independent investigators for the presence of the predefined criteria. The melanomas were subdivided according to their diameter into small melanomas, so-called micromelanomas (microM)—sized ≤ 5.0 mm and >5.0 mm, according to published definitions of small melanocytic lesions. The Triage Amalgamated Dermoscopic Algorithm (TADA) and the revisited 7-point checklist of dermoscopy (7-point) algorithm were chosen for the diagnostic feasibility. Odds ratios and corresponding 95% confidence limits (CL) were calculated using the logistic regression adjusted for age for the melanoma-specific dermoscopic structures, the dermoscopic patterns and the diagnostic feasibility of the 7-point checklist and TADA algorithms. The *p*-values of the results were corrected using the Bonferroni method. Results: In total, 106 patients with 109 melanomas, 50 sized ≤ 5.0 mm and 59 exceeding the diameter of 5.0 mm, were retrospectively analyzed. The prevalent general pattern of microM was the spitzoid one (48% vs. 11.86%, *p* = 0.0013). Furthermore, 40% of microM vs. 6.78% melanomas sized > 5.0 mm (*p* = 0.0023) did not present melanoma-specific patterns. The asymmetric multicomponent pattern was present in 64.41% melanomas sized > 5.0 mm and in 26.00% microM (*p* = 0.0034). The asymmetry of structures or colors was detected in 56% microM vs. 89.83% (*p* = 0.0020) and 56% microM and 94.92% (*p* = 0.000034) melanoma sized > 5.0 mm, respectively. The differences in frequency of the detected dermoscopic structures specific to melanomas revealed that microM are almost deprived of negative networks (*p* = 0.04), shiny white structures (*p* = 0.0027) and regression features (*p* = 0.00003). Neither prominent skin markings nor angulated lines were found in the entire study group. Out of the vascular structures, microM presented only dotted (32%) or polymorphous (28%) vessels, although more rarely than melanomas sized > 5.0 mm (66.1% *p* = 0.017 and 49% *p* > 0.05, respectively). The diagnostic feasibility revealed a score ≥ 3 of the 7-point algorithm (indicative for malignancy) in 60% microM and 98.31% melanomas sized > 5.0 mm (*p* = 0.000006). The TADA algorithm revealed melanoma-specific patterns in 64% microM and 96.61% > 5.0 mm-sized melanomas (*p* = 0.00006) and melanoma-specific structures in 72% and 91.53% (*p* > 0.05), respectively. Conclusion: In the dermoscopy, 40% of micromelanomas histopathologically staged as pTis and pT1a did not reveal melanoma-specific patterns. Among the general melanocytic patterns, the spitzoid one was the most frequently found in melanomas sized ≤ 5.0 mm. The 7-point checklist and TADA dermoscopic algorithms were helpful in the identification of the majority of melanomas sized ≤ 5.0 mm.

## 1. Introduction

In recent decades, dermoscopy has proved to be irreplaceable in the noninvasive diagnosis of melanoma, as well as being established for the secondary prevention of skin malignant neoplasms [1]. Cutaneous melanomas that have small diameters, also called micromelanomas (microM), have been reported with variable frequency within the range of 1.5–38.2%, depending on the threshold value obtained and the number of samples included in the study [2,3]. According to Akay et al., the smallest microM that has been detected by dermoscopic examination so far had a diameter of 0.9 × 0.6 mm [4].

The first publication describing the dermoscopic diagnosis of microM was carried out by Bono et al., 1999 [5]. To date, seventeen other studies have been published showing significant differences in the sizes of samples (8–123 microM; median 27.5) and methodologies [2,3,6,7,8,9,10,11,12,13,14,15,16,17,18,19,20] (Appendix A).

Unfortunately, the results regarding the diagnostic frequency of dermoscopic structures are difficult to compare, due to differences in the number of selected dermoscopic structures, chosen comparators and different ratios of in situ to invasive melanomas (pT1, >pT1) included. MicroM were analyzed among only five to six dermoscopic patterns in four studies (De Giorgi et al., 2012; Seidenari et al., 2014; Dika et al., 2017; Megaris et al., 2018) [12,14,17,19]. Five studies were conducted without the comparator group [8,9,12,16,19]. Studies conducted by Carli et al., 2003, Friedman et al., 2008, Abbasi et al., 2008, Pupelli et al., 2013, Drugge et al., 2018 and Campos-do-Carmo et al., 2021 chose melanocytic lesions as the comparator group [7,10,11,13,18,20]. Two of them analyzed microM with melanocytic nevi of the same diameter [13,20]. The remaining studies chose different types of melanomas as the comparator group [3,6,14,15,17]. The differences concerned the ratio of in situ and invasive melanomas (Pizzichetta et al., 2001; Helsing et al., 2004; Seidenari et al., 2014; Emiroglu et al., 2014) or clinical presentation–difficult-to-diagnose melanoma versus clinically evident melanoma (Dika et al., 2017) [3,6,14,15,17].

The diagnostic feasibility of dermoscopic algorithms in small-diameter melanomas was assessed by Bono et al., 2006 (Menzies method sensitivity 83%; specificity 69%), Seidenari et al., 2014 (7-point checklist and ABCD rule of Stoltz; data of sensitivity or specificity were not provided) Dika et al., 2017 (diagnostic sensitivity of pattern analysis 64.05%; 7-point checklist 61.08%; Menzies method 57.30%; ABCD rule of Stoltz 42.70%.) and Campos-do-Carmo et al., 2021 (modified ABC-point list algorithm sensitivity 61.8%; specificity 48.3%) [9,14,17,20]. Six studies (Bono et al., 1999, Carli et al., 2003, Bono et al., 2004 and 2006, Abbasi et al., 2008 and Campos-do-Carmo et al., 2021) compared the dermoscopic examination with the naked eye examination based on the clinical ABCDE rule with a wide range of sensitivity (36.6–83%) and specificity (38–91%) [5,7,8,9,11,20].

In the light of the presented published studies, our research was conducted on a uniform group of melanomas (in the context of histopathological invasiveness), where the diameter was the only cut-off criterion for investigation. The methodology of our study was highly detailed, as we evaluated according to all currently described patterns applied for melanocytic lesions. The reason for this decision was influenced by a suspicion that microM might simulate benign lesions [17,20,21,22,23,24,25,26,27]. As multiple tests were performed, the p-values of results were corrected using the Bonferroni method. We also assessed the diagnostic feasibility of the previously investigated and commonly known algorithm—the 7-point checklist—and a new one—the TADA algorithm—which has never been assessed before in microM [27,28,29].

Our study aimed to verify two hypotheses. The first hypothesis concerned the possibility of the dermoscopic differentiation of cutaneous melanomas histopathologically proven as in situ (pTis) and thin melanomas (pT1a) in terms of their diameter. The second hypothesis was aimed at the diagnostic feasibility of selected dermoscopic algorithms (7-point checklist and TADA algorithm) aiming to detect ≤5.0 mm-sized melanomas in the histopathological category as pTis and pT1a.

## 2. Methods

The research was a multicenter morphologic study conducted in 3 specialized dermatologic centers for skin cancer diagnosis and management in Poland. The databases of our centers were screened to identify eligible cases of all melanomas records stored in the videodermoscopic system between January 2012 and May 2020. Eligibility criteria were the histopathological diagnosis of thin (pT1a, Breslow thickness <0.8 mm) and in situ (pTis) melanomas located only within glabrous skin, as well as the availability of high-quality dermoscopic images of melanomas. The obtained and further analyzed databases included demographic (age and gender), anamnestic (risk factors and skin phototype according to Fitzpatrick) and clinical (anatomical location and diameter of melanoma) information, videodermoscopic images and histopathologic reports. The videodermoscopic records were received from Fotofinder HD 800 or Medicam 1000 (FotoFinder Systems GmbH, Bad Birnbach, Germany) or Mole Max (Derma Medical Systems Handels u. Entwicklungs GmbH Vienna, Austria) captured in polarized light, performed at the same 20-fold magnification.

All the histopathological specimens were primary evaluated by 2 independent pathomorphologists based on a standard procedure. The specimens were reassessed considering the TNM staging system/8th AJCC classification, but none of the primary results required an update [30].

All the images presented in this article are anonymous and are not identifiable, and the patients gave written informed consent to be photographed in their respective clinical centers.

### 2.1. The Selection Criteria

The primary inclusion criterion was the melanoma diameter with a cut-off ≤5 mm for the micromelanoma group and >5.0 mm for the comparative group. The diameters were measured automatically in two perpendicular axes by the calibrated videodermoscopes measuring programs. Consequently, the longest diameter of each lesion was considered as the final melanoma size for further evaluation. The choice of the diameter with a cut-off ≤5 mm for the micromelanoma was based on the analysis of the results of studies published in recent years, in which the criterion of 5–6 mm was most often applied [12,13,14,15,16].

The second inclusion criterion was the category in situ (pTis) and Breslow thickness melanoma <0.8 mm (pT1a) in the TNM staging system/8th AJCC classification of melanoma.

The third criterion was the location of the melanoma only on the glabrous skin (lesions on the face, scalp, acral, nail, and mucosal surfaces were excluded from the study).

### 2.2. Dermoscopic Assessment

Three expert dermoscopists with 20 years of experience reviewed and assessed melanoma videodermoscopic documentation performed at 20-fold magnification. The concordance observation rate was considered when 2 out of 3 dermatologists agreed on all dermoscopic assessments. The evaluation of the dermoscopic data included the following:General melanocytic lesion patterns defined as reticular; globular; peripheral globules with a central network; peripheral globules with central homogenous; reticular with central globules; homogenous tan; homogenous brown; homogenous blue; spitzoid: pseudopods or radial streaming/starburst pattern; globular; reticular; homogenous pink; homogenous with tiered globules; homogenous black; atypical; negative network with white shiny streaks; multicomponent symmetric; multicomponent asymmetric (≥3 structures: network, globules/dots/homogenous areas); nonspecific; two-component: reticular–homogenous, reticular–globular, globular–homogenous [22,23,24,25,26].Melanoma-specific patterns: asymmetric multicomponent pattern, structureless brown, structureless blue-black nodule, structureless pink/tan macule, not applicable [22,23,24,25,26].Melanoma-specific structures: atypical network, negative network, irregularly distributed dots/globules, irregularly distributed blotch, blue white veil (raised), flat blue white structureless area, black dots, blue-grey dots, multiple hyperpigmented areas; regression structures: peppering/granularity, erythema/vessels and scar-like depigmentation, prominent skin markings, polygons/angulated lines, atypical vessels: polymorphous, dotted, comma, serpentine (linear irregular), corkscrew, milky-red areas/globules, shiny white structures [22,23,24,25,26].Color type: light brown, dark brown, black, blue, red, white, grey.Architectural disorder: asymmetry of structures, asymmetry of colors.

### 2.3. The Diagnostic Feasibility

All melanomas were evaluated with the revised 7-point checklist of dermoscopy (7-point) cut-off ≥3 and the Triage Amalgamated Dermoscopic Algorithm (TADA) for pigmented and nonpigmented skin cancers, according to the description in original publications by Argenziano et al. and Rogers et al., respectively [27,28,29].

### 2.4. Statistical Analysis

The statistical analysis was performed using the Statistica version 10.0 software (StatSoft Inc., Tulsa, OK, USA) for Windows. The Pearson’s Chi2, Fisher exact two-tailed test and crosstabulation tables were applied for the frequency statistics. The nonparametric Wald–Wolfowitz test runs and Mann–Whitney U test for independent samples were applied for numerical data. Odds ratios and corresponding 95% confidence limits (CL) were calculated using logistic regression adjusted for age for the melanoma-specific dermoscopic structures, the dermoscopic patterns and the diagnostic feasibility of the selected algorithms. As multiple tests were performed, the *p*-values of the results were corrected using the Bonferroni method. *p* < 0.05 was considered statistically significant.

## 3. Results

### 3.1. Characteristics of The Study Population

The study population consisted of 106 patients with a profound medical history of 109 melanomas. All the subjects complied with the inclusion criteria: 48 patients diagnosed with 50 micromelanomas and 58 patients diagnosed with 59 > 5 mm-sized melanomas assigned to the comparative group. The epidemiological, anamnestic, clinical and histopathologic findings are presented in Table 1.

The female gender dominated by 60% versus 40% of males in both groups (*p* > 0.05). Despite the comparable age range, patients with micromelanomas were significantly younger (*p* = 0.018), as shown in the mean and median value regarding the age variable—43.27 years and 39.5 years versus 51.51 years and 51.50 years, respectively. The location of the melanoma was similar in both groups (*p* > 0.05)—on the trunk, the thigh, and the upper limb. The risk analysis revealed the predominant 2nd skin phototype (*p* > 0.05), multiple nevi regarding 29.16% of micromelanoma patients, as well as 56.89% of patients in the comparative group (*p* < 0.005). The severe photodamaged skin variable accounted for 25% and 50%, respectively (*p* < 0.01). There were few patients with a concomitant or previous history of melanoma. The ratio of melanomas in situ and thin ones in the study groups revealed statistically significant differences (*p* < 0.00003), as 49.15% of melanomas sized >5 mm versus 12% microM were stage pT1a.

### 3.2. Dermoscopic Assessment

The evaluation of general melanocytic lesion patterns included 22 patterns describing benign, spitzoid, and atypical lesions (Table 2, Figure 1) [22,23,24,25,26]. Figure 1 presents the revealed spitzoid patterns of microM. Additionally, all the lesions were evaluated based on the four types of melanoma—specific patterns (Table 2) [22,23,24,25].

The most frequently appearing general pattern in the micromelanoma group was the spitzoid one (48% vs. 11.86%, *p* = 0.0004, a*p* = 0.0013) with predominant starburst, atypical and homogenous with tiered globules subtypes. The most striking pattern in the >5 mm-sized melanomas group was the general asymmetric multicomponent pattern, revealed in 64.41%, in comparison to 26.00% in the microM group (*p* = 0.0001, a*p* = 0.0034). It is important that 40% of the microM versus 6.78% in the comparative group (*p* = 0.00007, a*p* = 0.0023) did not present melanoma-specific patterns. Furthermore, 42% of the microM versus 83.5% of the comparative group (*p* = 0.00001, a*p* = 0.00034) manifested melanoma-specific multicomponent asymmetric patterns. The architectural disorder observed as the asymmetry of structure and color distribution was present in 56% of microM compared to 89.83% (*p* < 0.000001, a*p* = 0.000034) for colors and 94.92% (*p* = 0.00006, a*p* = 0.002) for structures in the >5 mm-sized melanomas group (Table 2).

The multivariate analysis with logistic regression of the following variables remained in the adjusted for age mode: spitzoid (*p* = 0.00056, a*p* = 0.019, OR—0.16, 95% CL—0.06–0.45, SE—0.50) and muliticomponent asymmetric patterns (*p* = 0.00099, a*p* = 0.033, OR—4.34, 95% CL—1.83–10.25, SE—0.43) for the general melanocytic patterns; the multicomponent asymmetric (*p* = 0.00018, a*p* = 0.006, OR—5.90, 95% CL—2.38–14.62, SE—0.45) and melanoma nonspecific (not applicable) (*p* = 0.00092, a*p* = 0.031, OR—0.12, 95% CL—0.03–0.4, SE—0.60) patterns for the melanoma-specific patterns; colors (*p* = 0.00012, a*p* = 0.004, OR—14.42, 95% CL—3.81–54.60, SE—0.67) and structures (0.00048, a*p* = 0.016, OR—6.70, 95% CL—2.35–19.12, SE—0.52) for the architectural disorder (Table 3).

The comparison of the frequency of melanoma-specific structures between the study groups revealed that microM are almost deprived of negative networks (*p* = 0.0012, a*p* = 0.04), shiny white structures (*p* = 0.00008, a*p* = 0.0027) and regression features—both granularity/peppering subtype (*p* = 0.00001, a*p* = 0.00034) and erythema/vessels/scar-like depigmentation type (*p* < 0.000001, a*p* = 0.000034) (Table 2). Furthermore, neither prominent skin markings nor angulated lines were not found in the entire study group. Although microM featured irregularly distributed dots/globules (48%), streaks/pseudopods structures (46%), an atypical network (40%) and blue-grey dots (24%), the only statistically significant difference to the comparative group was found for the atypical blotch structure (30%, *p* < 0.000001, a*p* = 0.000034) (Table 2). Out of the vascular structures, the frequency statistics revealed that microM presented almost uniquely dotted (32%) or polymorphous (28%) vessels, although more rarely than melanomas sized >5.0 mm (66.1% *p* = 0.0005, a*p* = 0.017 and 49% *p* = 0.03, a*p* > 0.05, respectively) (Table 2).

The multivariate analysis with the logistic regression adjusted for age of the melanoma-specific structures maintained the variables for regression structures of any type (*p* = 0.0000009, a*p* = 0.00003, OR—12.33, 95% CL—4.74–32.09), shiny white structures (*p* = 0.00083, a*p* = 0.028, OR—15.59, 95% CL—3.2–76.04), atypical blotches (*p* <0.00004, a*p* = 0.0013, OR—6.60, 95% CL—2.75–15.85), dotted vessels (*p* = 0.00070, a*p* = 0.023, OR—4.5, 95% CL—1.91–10.62), serpentine/linear irregular vessels (*p* = 0.000029, a*p* = 0.0009, OR—13.04, 95% CL—3.85–44.14) and comma/curved vessels (*p* = 0.0011, a*p* = 0.037, OR—14.61, 95% CL—2.97–71.72) (Table 4).

The analysis of detected colors revealed statistically significant differences in Pearson’s Chi2 test within the study group (*p* = 0.00009) regarding color: white (*p* < 0.000001), grey (*p* = 0.00025), red (*p* = 0.0063) and light brown (*p* = 0.042) based on the Fisher exact two-tailed test. The most frequently found colors in both groups as well as in over 40% of micromelanomas were light and dark brown and red. The comparative group also featured white and grey colors. The Wald–Wolfowitz runs test proved the statistical significance of the coexistence of two–three colors (median 3) in the micromelanoma group in comparison to 3–5 colors (median 4) in the comparative group. The logistic regression adjusted for age for colors revealed a statistically significant *p*-value for red (*p* = 0.0025; a*p* = 0.08 SE—0.43, OR—3.81, 95% CL—1.61–9.00), white (*p* = 0.000024; a*p* = 0.0008 SE—0.59, OR—13.72, 95% CL—4.23–44.51) and grey (*p* = 0.00028; a*p* = 0.009 SE—0.44, OR—5.26, 95% CL—2.18–12.66) colors.

The criteria of the 7-point cut-off at ≥ three scores complied with 60% of the micromelanoma group and 98.31% of melanomas >5 mm in size (*p* < 0.000001, a*p* = 0.000006) (Table 5). The additional criteria of this algorithm—named the 7 rules not to miss melanoma incognito—were applied in 72% and 22.03%, respectively (*p* < 0.000001, a*p* = 0.000006). The application of the TADA algorithm revealed melanoma-specific patterns in 64% of the micromelanomas and in 96.6% of the comparative group (*p* = 0.00001, a*p* = 0.00006) (Table 5). Furthermore, it displayed melanoma-specific features in 72% and 91.5%, respectively (*p* = 0.0105, a*p* > 0.05). According to the final summary, 3 (6%) microM did not comply with any criteria for melanoma of TADA, and 6 (12%) did not comply with the 7-point checklist, also including an additional 7 points. Both diagnostic algorithms constituted 100% in the >5 mm-sized melanoma group versus 88% in the 7-point check list and 94% in TADA for microM, both of which were not statistically significant. Therefore, the diagnostic feasibility of the 7-point checklist and TADA algorithms proved to be equally efficient in the detection of melanomas pTis and pT1a, regardless of their size (*p* > 0.05), when both diagnostic steps (A and or B) were applied (Table 5).

## 4. Discussion

The analysis of the published studies revealed important discrepancies regarding the sample size, the diameter of lesions, the proportion of in situ versus invasive melanomas and the presence and type of the comparator, which might have had an impact on the obtained results [2,3,5,6,7,8,9,10,11,12,13,14,15,16,17,18,19,20]. An essential remark in the context of our study is that the dermoscopic structures and patterns, as well as the diagnostic algorithms applied by many authors, are difficult to compare due to high variability.

The most remarkable conclusion of our study was that one of the spitzoid patterns might be the first dermoscopic symptom of microM (48% vs. 11.86%, *p* = 0.0004, a*p* = 0.0013), although it has not been reported in this frequency previously [14,19]. De Giorgi et al. revealed mainly unspecific (76.5% versus 39.1%, *p* < 0.001) and reticular (17.7% versus 33%) patterns in 34 micromelanomas sized <6 mm in comparison to melanocytic nevi of the same diameter in contrast to single cases in microM of globular (2.9% versus 14.5%), starburst (spitzoid) (2.9% versus 4.4%) and homogenous (0 versus 9%) patterns [12]. Seidenari et al. found six dermoscopic patterns in a group of 22 micromelanomas sized ≤4 mm: multicomponent 32%, bicomponent 27%, spitzoid 18%, reticular 14%, globular and island 5% [14]. Megaris et al. reported different frequencies of global patterns in melanomas with diameters up to 5 mm, including reticular, 57.7%; structureless, 26.9%; mixed (globular and reticular), 11.5%; starburst (spitzoid), 3.8%; and none for globular and multicomponent ones [19]. Our conclusion is not surprising, as the dermoscopic spitzoid pattern is rarely confirmed by the spitzoid melanoma diagnosis in the histopathological report [31,32,33,34,35]. In our study, no micromelanomas were classified as spitzoid melanomas the in histological report. The most important consideration, according to studies reported by Lallas et al., is that the probability of a dermoscopically symmetric spitzoid-like lesion being a melanoma depends on the patient’s age: it is extremely low before puberty and gradually increases afterwards—50% after the age of 50 years [34,35].

Another important remark of our study was the finding that 40% of the microM versus 6.78% of larger melanomas (*p* = 0.00007, a*p* = 0.0023) did not present melanoma-specific patterns, despite their in situ or early invasive stage (pT1a). The study of De Giorgi et al. revealed an unspecific pattern in 76.5% of microM versus 39.1% of melanocytic nevi of the same size (*p* < 0.001) [12]. The results prove the diagnostic difficulties of the micromelanoma, as a potential imitator of the melanocytic nevi, which was reported in respect to in situ micromelanomas and to melanomas [20,21,36,37,38,39].

The most widespread discrepancy between the referenced studies can be observed in dermoscopic structures and their significance for melanomas with small diameters, which have led the authors to divergent conclusions describing the morphology of mircoM [2,3,5,6,7,8,9,10,11,12,13,14,15,16,17,18,19,20]. In the recent publication of Campos-do-Carmo G et al., 2021 comparing the dermoscopic structures of 123 micromelanomas of less than 6 mm in diameter (the largest sample among the published studies) with 358 melanocytic nevi of the same diameter, the independent variables associated with the diagnosis of melanoma adjusted for age, gender and location were as follows: streaks (adjusted odds ratio (AOR)—2.5; 95% CI 1.3–4.7; *p* = 0.006) and the presence of a structureless area (AOR—2.2, 95% CI 1.2–4.0, *p* = 0.011), conversely to a symmetric typical pigment network, which was a protection variable (AOR—0.4, 95% 0.7–0.9, *p* = 0.040) [20]. The features of micromelanomas revealed by De Giorgi et al., Seidenari et al., Emiroglu et al. and Megaris et al. were the presence of asymmetric structures (36–79.3%), atypical networks (42.3–77%), irregular dots/globules (55–88.4%), irregular streaks (34.6–58.6%), irregular blotches (38.4–79.3%), regression (7.6–52.9%), irregular pigmentation (29.1–88.4%), atypical vessels (3.8–13.8%) and blue-white (grey) veils (3.8–65.5%) [12,14,15,19]. It is interesting that some structures were indicated only by single studies. Negative networks (11.5%), prominent skin markings (11.5%) and angulated lines (46.2%) were only found in the study of Megaris et al., whereas a variety of colors (72.4%) and milky red areas (24.1%) were found in the study of Emiroglu et al. [15,19]. The results of our study confirmed the data regarding the architectural disorder, multicomponent patterns and dermoscopic structures reported in the studies of De Giorgi et al., Seidenari et al., Emiroglu et al. and Megaris et al. with discrepancies in their frequencies [12,14,15,19]. Our distinctive conclusion was that microM are almost deprived of negative networks (*p* = 0.04), shiny white structures (*p* = 0.0027) and regression features (*p* = 0.00003). Out of the vascular structures, microM contained only dotted (32%) or polymorphous (28%) vessels, although more rarely than melanomas sized >5.0 mm (66.1% *p* = 0.017 and 49% *p* > 0.05, respectively).

The study of Dika et al. is the only one to assess the diagnostic feasibility of four diagnostic algorithms [17]. The sensitivity of the modified pattern analysis, ABCD rule of dermoscopy, 7-point check list of Argenziano and Menzies method for thin melanomas of the lower limbs was lower than in previous studies and reached 64.05%, 42.70%, 61.80% and 57.30%, respectively [17]. Pupelli et al. analyzed the 7-point check list and obtained a positive diagnose (score ≥ 3 points) in 22/24 of melanomas of under 5 mm in diameter [13]. The diagnostic feasibility of the revised 7-point checklist in our study revealed a sensitivity of 60% and 98.31% in the micromelanoma group and the >5 mm-sized melanomas, respectively (*p* < 0.000001, a*p* = 0.000006) (Table 5). The additional criteria, i.e., 7 rules to avoid missing melanoma, were applied in 72% of the <5 mm-sized melanoma group versus 22.03% of the larger ones (*p* < 0.000001, a*p* = 0.000006) and helped in the diagnosis of another 14 cases. The melanomas that complied with one or both of these criteria reached 88% in the micromelanoma group, indicating that only six of the micromelanomas were diagnosed as a false negative. The TADA algorithm has never been evaluated in the micromelanoma group before. The results showed that the melanoma-specific pattern was present in 64% of the microM group and 96.6% of the comparative group (*p* = 0.00001, a*p* = 0.00006), and the melanoma-specific features in 72% (*p* > 0.05), which finally leads to three microM cases that remained undiagnosed.

The color of melanoma varies depending on histological features and invasiveness [40]. The study of Emiroglu et al. reported the presence of at least three colors in 78.8% of melanomas [15]. Bono et al. underlined that most micromelanomas displayed black/dark brown or uniform color [5,8]. Our study revealed that brown and red colors were the most common in micromelanomas. The Wald–Wolfowitz runs test showed statistical significance for the coexistence of two to three colors in the microM group and the coexistence of 3–5 colors (median 4) in the comparative group.

The differences in the patients’ age and gender ratios were inconsistent within the range of analyzed studies [2,3,5,6,7,8,9,10,11,12,13,14,15,16,17,18,19,20]. Seidenari et al. showed that the younger population was diagnosed with tiny melanomas—mean age 42.18—compared to melanomas >4 mm in diameter 55.52, and with a slight female prevalence (59%) [14]. Fernandez et al. showed similar conclusions based on the analysis of pathological reports (average age 52.45 years in microM versus 59.16 years in melanomas >6 mm, *p* < 0.002) [2]. Our study confirmed these findings, as the median of age in micromelanomas was 39.5 years, compared to 51.5 years in the comparative group (*p* = 0.018), with the prevalence of female gender equally represented in both groups by approximately 60% (*p* > 0.05). It seems that further education campaigns regarding the primary and secondary prevention of melanoma should be directed toward the male population, which less frequently attends dermoscopic screening appointments.

The identification of micromelanomas is a challenge, especially in the case of high-risk patients. This diagnostic problem is commonly resolved by the application of short- or long-term digital dermoscopy monitoring [12,14,16,17,18,36,41,42,43,44]. Our study explored the dermoscopic morphology of micromelanomas among >5 mm-sized melanomas, stages pTis and pT1a, indicating that the application of the 7-point checklist including the 7 rules not to miss melanoma incognito or the TADA algorithms might help to identify approximately 90% of micromelanomas.

## 5. Conclusions

The dermoscopy of micromelanomas staged as pTis and pT1a did not reveal a melanoma-specific pattern in 40% of the microM (*p* = 0.00007, a*p* = 0.0023). Among the general melanocytic patterns, the spitzoid one was the most frequently found in microM. Knowledge of these dermoscopic features will bring the clinician closer to an early diagnosis of melanomas with a diameter of 5 mm or less. The 7-point checklist and TADA dermoscopic algorithms were helpful in the diagnosis of the majority of micromelanomas.

Further studies based on a convolutional neural network may systematize the dermoscopic patterns and structures of different types of melanomas, and simultaneously analyze relationships with multiple variables, such as clinical factors (skin phototype, number of melanocytic nevi and lentiginosis), anamnestic data on previous skin neoplasms or patients’ characteristics [45,46].

Limitations of the study: The first limitation of our study is the retrospective analysis and lack of comparison with the benign melanocytic lesions, including small-diameter nevi. The bias of the study was the selective inclusion of melanomas with fully archived videodermoscopic records. Another bias of the study was the profiles of patients from dermatologic centers, where small-diameter melanomas predominate, as opposed to oncology centers, where patients usually present at more advanced clinical and histopathologic stages. The material was exclusively obtained from dermatological centers, where the same methods of dermoscopic examination and archiving were used; therefore, no differences were found in the analyzed patient population.

## Figures and Tables

**Figure 1 cancers-13-06095-f001:**
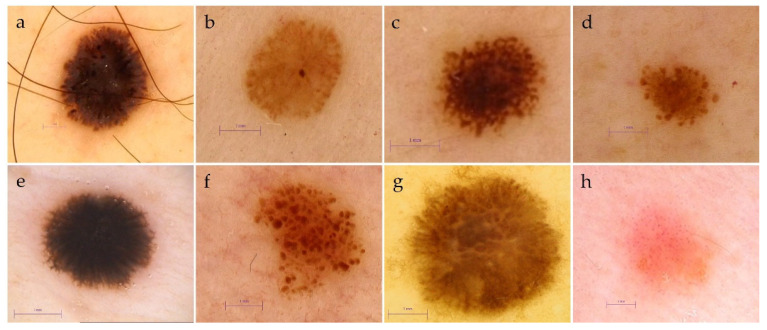
The spitzoid patterns of micromelanomas revealed: (**a**–**c**) starburst pattern with pseudopods or radial streaming, (**d**) homogenous with tiered globules, (**e**) reticular, (**f**) globular, (**g**) atypical, (**h**) homogenous pink with multiple dotted vessels, described in dermoscopic assessment (magnification 20–70×, Fotofinder HD 800 or Medicam 1000 (FotoFinder Systems GmbH, Bad Birnbach, Germany).

**Table 1 cancers-13-06095-t001:** Demographic, clinical and histopathologic data in the analyzed groups of patients. Small sample sizes of skin phototype (I, III, IV, V), previous/concomitant melanoma or nonmelanoma skin cancer history and some locations of melanoma prevented the calculation of frequency differences due to insufficient statistical power (data of *p*-value not provided). NS—not statistically significant.

Variable	Micromelanoma Group	Sized in Diameter >5.0 mm Melanoma (Comparative Group)	*p*-Value
Number of patients (*n*)	48	58	
Gender Female (F), male (M)	F29 (60.42%) M 19 (39.58%)	F 35 (60.34%) M 23 (39.66%)	NS
Age	(21–82) y	(18–89)	0.018
Mean	43.27 y	51.51 y	
Median	39.5 y	51.50 y	
SD±	14.23	18.66	
Diameter of melanomas (mm) ≤2 2.1–3.0 3.1–4.0 4.1–5.0 5.1–6.0 6.1–7.0 ≥7.1 Mean diameter axis #1 Median diameter axis #1 Mean diameter axis #2 Median diameter axis #2	8/50 (16%) 9/50 (18%) 12/50 (24%) 21/50 (42%) - - - 3.26 3.00 3.65 4.00	- - - - 10/59 (16.94%) 1/59 (1.69%) 48/59 (81.35%) 13.90 10.00 9.83 7.00	
Histopathological diagnosis based on TNM staging Extrafacial lentigo maligna Melanoma in situ (pTis) Thin melanoma (pT1a) LM + Melanoma pTis vs. Melanoma pT1a	1/50 (2%) 43/50 (86%) 6/50 (12%) 44/50 (88%) 6/50 (12%)	2/59 (3.38%) 28/59 (47.45%) 29/59 (49.15%) 30/59 (50.85) 29/59 (49.15%)	<0.00003
Skin phototype according to Fitzpatrick	I—3/48 (6.25%)	I—10/58 (17.24%)	
	II—44/48 (91.66%)	II—46/58 (79.31%)	NS
	III—1/48 (2.08%)	III—2/58 (3.44%)	
	IV—0/48 (0.00%)	IV—0/58 (0.00%)	
	V—0/48 (0.00%)	V—0/58 (0.00%)	
Location of melanoma Head/neck Trunk Arm Forearm Thigh Lower leg Hand (dorsal area) Foot (dorsal area)	1/50 (2%) 20/50 (40%) 5/50 (10%) 5/50 (10%) 16/50 (32%) 2/50 (4%) 0/50 (0%) 1/50 (2%)	0/59 (0.00%) 32/59 (54.23%) 6/59 (10.16%) 7/59 (11.86%) 11/59 (18.64%) 2/59 (3.38%) 0/59 (0%) 1/59 (1.69%)	NS NS
Multiple nevi or atypical nevus syndrome			<0.005
Yes	14/48 (29.16%)	33/58 (56.89%)	
No	34/48 (70.83%)	25/58 (43.10%)	
Severe photodamaged skin			<0.01
Yes	12/48 (25%)	29/58 (50%)	
No	36/48 (75%)	29/58 (50%)	
Previous melanoma in the history	4/48	2/58	
Concomitant melanoma	1/48 (on trunk)	4/58	
Melanoma in family history	1/48	1/58	
Previous or concomitant nonmelanoma skin cancers	2/48	3/58	

**Table 2 cancers-13-06095-t002:** Comparison of the dermoscopic patterns (general melanocytic lesions pattern and melanoma specific patterns) [22,23,24,25,26], architectural disorder, and dermoscopic structures [22,23,24,25,26] detected in the analyzed groups. Small sample sizes of the selected patterns prevented the calculation of frequency differences due to insufficient statistical power (NC—not calculated). The statistical significance of *p* < 0.05 was considered. NS—statistically not significant.

Dermoscopic assessment	Total Number of Melanomas *n* = 109	Micromelanoma Group *n* = 50	Sized in Diameter >5.0 mm Melanoma (Comparative Group) *n* = 59	Fisher Exact Test, Two-Tailed	
				*p*-Value	Adjusted *p*-Value (Bonferroni)
**General melanocytic patterns**	**Pearson Chi2 test *p* = 0.00028**				
Reticular	12 (11.01%)	6 (12%)	6 (10.17%)	NC	
Globular	0	0	0		
Peripheral globules with central network	0	0	0		
Peripheral globules with central homogenous area	0	0	0		
Reticular with central globules	0	0	0		
Homogenous tan	0	0	0		
Homogenous brown	4 (3.67%)	4 (8%)	0	NC	
Homogenous blue	0	0	0		
Spitzoid—pseudopods or radial streaming/starburst pattern	6 (5.50%)	6 (12%)	0 (0.00%)	NC	
Spitzoid—globular	2 (1.83%)	2 (4%)	0 (0.00%)	NC	
Spitzoid—homogenous pink	6 (5.50%)	2 (4%)	4 (6.78%)	NC	
Spitzoid—homogenous with tiered globules	6 (5.50%)	5 (10%)	1 (1.69%)	NC	
Spitzoid—reticular	4 (3.67%)	3 (6%)	1 (1.69%)	NC	
Spitzoid—homogenous black	0	0	0		
Spitzoid—atypical	7 (6.42%)	6 (12%)	1 (1.69%)	NC	
Spitzoid—negative network with white shiny structures	0	0	0		
Spitzoid—any type	31 (28.44%)	24 (48%)	7 (11.86%)	**0.00004**	**0.0013**
Multicomponent symmetric	0	0	0		
Multicomponent asymmetric— ≥3 structures (network, globules/dots/homogenous areas)	51 (46.79%)	13 (26%)	38 (64.41%)	**0.0001**	**0.0034**
Nonspecific	5 (4.59%)	2 (4%)	3 (5.08%)	NC	
Two-component—reticular—homogenous	5 (4.59%)	1 (2%)	4 (6.78%)	NC	
Two-component—reticular—globular	1 (0.92%)	0 (0.00%)	1 (1.69%)	NC	
Two-component—globular—homogenous	0	0	0		
**Melanoma specific patterns**	Pearson Chi2 test *p* < 0.000001				
Multicomponent pattern asymmetric	70 (64.22%)	21 (42%)	49 (83.5%)	**0.00001**	**0.00034**
Structureless brown	6 (5.52%)	6 (12%)	0	0.0078	NS
Structureless blue-black nodule	0	0	0		
Structureless pink/tan macule	9 (8.28%)	3 (6%)	6 (10.14%)	NS	NS
Not applicable	24 (22.02%)	20 (40%)	4 (6.78%)	**0.00007**	**0.0023**
**Architectural disorder**					
Asymmetry of structures —present	81 (74.31%)	28 (56%)	53 (89.83%)	**0.00006**	**0.002**
Asymmetry of colors —present	84 (77.06%)	28 (56%)	56 (94.92%)	**<0.000001**	**0.000034**
**Melanoma specific structures**					
Polymorphous atypical vessels—present	43 (39.45%)	14 (28%)	29 (49.15%)	0.03	NS
Dotted vessels—present	55 (50.46%)	16 (32%)	39 (66.1%)	**0.0005**	**0.017**
Comma vessels—absent	87 (79.82%)	48 (96%)	39 (66.10%)	**0.00008**	**0.0027**
Serpentine vessels—absent	76 (69.72%)	46 (92%)	30 (50.85%)	**<0.000001**	**0.000034**
Corkscrew vessels—absent	103 (94.5%)	49 (98%)	54 (91.53%)	NS	NS
Milky red areas or globules—absent	90 (82.57%)	47 (94%)	43 (72.88%)	0.0046	NS
Irregular dots/globules—present	58 (53.21%)	24 (48%)	34 (57.63%)	NS	NS
Black dots—absent	89 (81.65%)	40 (80%)	49 (83.05%)	NS	NS
Blue-grey dots—present	35 (32.11%)	12 (24%)	23 (38.98%)	NS	NS
Irregular blotches—present	59 (54.13%)	15 (30%)	44 (74.58%)	**<0.000001**	**<0.000034**
Irregular streaks/pseudopods—present	35 (32.11%)	23 (46%)	12 (20.34%)	0.007	NS
Atypical network—present	62 (56.88%)	20 (40%)	42 (71.19%)	0.0017	NS
Negative network—absent	88 (80.73%)	47 (94%)	41 (69.49%)	**0.0012**	**0.040**
Shiny white structures—absent	87 (79.82%)	48 (96%)	39 (66.1%)	**0.00008**	**0.0027**
Regression granularity/peppering—absent	78 (71.56%)	46 (92%)	32 (54.24%)	**0.00001**	**0.00034**
Regression erythema/vessels/scar-like depigmentation—absent	72 (66.05%)	45 (90%)	27 (45.76%)	**<0.000001**	**0.000034**
Blue-white structureless areas (raised)—absent	102 (93.58%)	49 (98%)	53 (89.83%)	NS	NS
Blue—white veil (flat)—absent	103 (94.5%)	50 (100%)	53 (89.83%)	0.03	NS
Multiple small hyperpigmented areas —absent	91 (83.49%)	43 (86%)	48 (81.36%)	NS	NS
Polygones/angulated lines	0	0	0		
Prominent skin markings	0	0	0		

**Table 3 cancers-13-06095-t003:** The logistic regression of dermoscopic patterns adjusted for age in the study group. The statistical significance of *p* < 0.05 was considered. OR—odds ratio; CL—confidence limits; SE—standard error; NS—statistically not significant.

Dermoscopic Patterns	Logistic Regression Adjusted for Age			
	*p*-Value	Adjusted *p*-Value (Bonferroni)	OR (95% CL)	SE
General pattern— spitzoid any type	**0.00056**	**0.019**	0.16 0.06–0.45	0.5
General pattern— Multicomponent asymmetric	**0.00099**	**0.033**	4.34 1.83–10.25	0.43
Melanoma specific pattern— Multicomponent asymmetric	**0.00018**	**0.006**	5.9 2.38–14.62	0.45
Melanoma specific pattern— Not applicable	**0.00092**	**0.031**	0.12 0.03–0.4	0.6
Asymmetry— of structures	**0.00048**	**0.016**	6.7 2.35–19.12	0.52
Asymmetry— of colors	**0.00012**	**0.004**	14.42 3.81–54.60	0.67

**Table 4 cancers-13-06095-t004:** The logistic regression of dermoscopic structures adjusted for age in the study group. The statistical significance of *p* < 0.05 was considered. OR—odds ratio; CL—confidence limits; SE—standard error; NS—statistically not significant.

Melanoma-Specific Structures	Logistic Regression Adjusted for Age		
Melanoma Specific Structures	*p*-Value	Adjusted *p*-Value (Bonferroni)	OR (95% CL) SE
Black dots	0.689	NS	0.81 (0.29–2.2) 0.51
Blue-grey dots	0.1	NS	2.07 (0.85–4.98) 0.44
Dots/globules irregularly distributed	0.31	NS	1.5 (0.67–3.31) 0.4
Streaks/pseudopods irregularly distributed	0.012	NS	0.32 (0.13–0.78) 0.44
Atypical blotches	**<0.00004**	**0.0013**	6.60 (2.75–15.85) 0.44
Multiple small hyperpigmented areas	0.57	NS	1.35 (0.46–4.00) 0.54
Atypical network	0.0029	NS	3.58 (1.55–8.23) 0.41
Negative network	0.0088	NS	5.96 (1.58–22.50) 0.66
Blue-white veil	0.13	NS	5.38 (0.58–49.35) 1.11
Shiny white structures	**0.00083**	**0.028**	15.59 (3.2–76.04) 0.79
Regression granularity/peppering	**<0.00049**	**0.0166**	8.42 (2.59–27.30) 0.59
Regression erythema/vessels/scar-like depigmentation	**<0.000045**	**0.0015**	10.51 (3.50–31.49) 0.55
Blue-white veil (flat)	9.97	NS	6.89 (0.00–) 8.56
Regression any type	**0.0000009**	**0.00003**	12.33 (4.74–32.09) 0.48
Vessels any type	0.0089	NS	3.13 (1.34–7.35) 0.42
Polymorphous vessels	0.0296	NS	2.54 (1.09–5.91) 0.42
Dotted vessels	**0.0007**	**0.023**	4.5 (1.91–10.62) 0.43
Comma/curved vessels	**0.0011**	**0.037**	14.61 (2.97–71.72) 0.80
Serpentine/linear irregular vessels	**0.000029**	**0.0009**	13.04 (3.85–44.14) 0.61
Corkscrew vessels	0.24	NS	3.84 (0.39–37.26) 1.14
Milky red areas/globules	0.013	NS	5.55 (1.43–21.50) 0.68

**Table 5 cancers-13-06095-t005:** The diagnostic feasibility of the selected algorithms proved to be equally efficient in detection of melanomas pTia and pT1a regardless of the size. The statistical significance of *p* < 0.05 was considered. SE—standard error; CL—confidence limits; NS—statistically not significant.

Diagnostic Algorithms	Frequency Statistic (Fisher Exact Two-Tailed Test)				Logistic Regression Adjusted for Age	
	Total	Micro Melanoma *n* = 50	Sized in Diameter >5.0 mm Melanoma (Comparative Group) *n* = 59	*p*-Value	*p*-Value SE	OR (95%CL)
				Adjusted *p*-Value (Bonferroni)		
7-point checklist						
A/Total score ≥ 3	88 (80.73%)	30 (60%)	58 (98.31%)	**<0.000001**	0.00045 1.08	50.33 (5.89–429.95)
				**0.000006**		
B/7 rules to avoid missing melanoma—applied	49 (44.95%)	36 (72%)	13 (22.03%)	**<0.000001**	0.000006 0.45	0.11 (0.04–0.28)
				**0.000006**		
Indication for melanoma—presence of criteria A or B	103 (94.49%)	44 (88%)	59 (100%)	>0.05	NC	NC
				NS		
TADA algorithm						
A/Melanoma specific pattern	89 (81.65%)	32 (64%)	57 (96.61%)	**0.00001**	0.00044 0.80	18.47 (3.74–91.10)
				**0.00006**		
B/Melanoma specific features	90 (82.57%)	36 (72%)	54 (91.53%)	**0.0105**	0.018 0.57	3.94 (1.26–12.27)
				**NS**		
Indication for melanoma—presence of criteria A or B	106 (97.25%)	47 (94%)	59 (100%)	>0.05	NC	NC
				NS		

## Data Availability

All data used in the study are available from the corresponding author upon request.

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
