# Peer review of "Dermoscopy of Small Diameter Melanomas with the Diagnostic Feasibility of Selected Algorithms—A Clinical Retrospective Multicenter Study"

_cancers, 2021, doi:10.3390/cancers13236095_

Round 1

Reviewer 1 Report

Citations are sometimes put before the end of the sentence (e.g. line 78) or after (e.g. line 80). Please do it in a uniform way. 

Was there Bonferroni correction for multiple testing? It seems that a large number of tests has been performed whith a relative small sample size.

How were the specimen selected? Could there be some selection bias? 

Was there a difference in the results between the individual centers?

Author Response

(1) Citations are sometimes put before the end of the sentence (e.g. line 78) or after (e.g. line 80). Please do it in a uniform way.
- Location of citations was revised and uniformly corrected.
(2) Was there Bonferroni correction for multiple testing? It seems that a large number of tests has been performed with a relative small sample size.
- The statistical analysis was thoroughly revised and all p-values of results were corrected using Bonferroni method. The description of the statistical analysis used in the study was revised and corrected as cited below.
The statistical analysis was performed using the Statistica version 10.0 software for Windows. The Pearson's Chi2, Fisher exact two-tailed test and the crosstabulation tables were applied for the fre-quency statistics. The nonparametric Wald-Wolfowitz test runs and Mann-Whitney U test for in-dependent samples were applied for numerical data. Odds ratios and corresponding 95% confi-dence limits (CL) were calculated by the logistic regression adjusted for age for the melanoma spe-cific dermoscopic structures, the dermoscopic patterns and for the diagnostic feasibility of selected
Cancers 2021, 13, x FOR PEER REVIEW 2 of 15
algorithms. Because multiple tests were performed, the p-values of results were corrected using Bonferroni method. The statistical significance of p < 0.05 was considered.
(3) How were the specimen selected? Could there be some selection bias? Was there a difference in the results between the individual centers?
- The bias of the study could be the selective inclusion of melanomas with fully archived vide-odermoscopic records – what was mention in the Limitations of the study.
- The material came from exclusively dermatological centers, where the same methods of dermo-scopic examination and archiving were used, therefore no differences were found in the patient population among these centers. As the dermatological centers (in contrast to the oncological ones) diagnose mainly early melanomas, the respective information was added to the Limitations of the study.
Limitations of the study: The first limitation of our study is the retrospective analysis and lack of the comparison with the benign melanocytic lesions including small diameter nevi. The bias of the study was the selective inclusion of melanomas with fully archived videodermoscopic records. Other bias of the study was the profile of patients from dermatologic centres, where the small di-ameter melanomas predominate, as opposed to oncology centres, where patients usually present at more advanced clinical and histopathologic stages. The material came from exclusively dermato-logical centres where the same methods of demoscopic examination and archiving were used there-fore no differences were found in the analyzed patient population.
(4) Does the introduction provide sufficient background and include all relevant references? Must be improved
- The introduction was completely revised and the background with summary of all 18 published studies was added to the text. We have described the design and general results of those studies. In the discussion we have reviewed their results in detail.
In recent decades, dermoscopy has proved to be irreplaceable in the non-invasive diagnosis of mel-anoma, as well as established in skin malignant neoplasms’ secondary prevention [1]. Small-sized in diameter cutaneous melanomas, also called micromelanomas (microM), were reported with var-iable frequency within the range of 1.5-38.2%, depending on the threshold value obtained and the number of samples included in the study [2,3]. According to Akay et al., the smallest microM that has been detected by dermoscopic examination so far had a diameter of 0.9 x 0.6 mm [4].
The first publication describing the dermoscopic diagnosis of microM was carried out by Bono et al. 1999 [5]. So far, seventeen other studies have been published differing significantly in the size of the sample (8-123 microM; median 27.5) and methodology [2,3,6-20] (Supplementary Table).

Unfortunately the results regarding the diagnostic frequency of dermoscopic structures are difficult to compare, due to differences in the number of selected dermoscopic structures, chosen compara-tor and different ratio of in situ to invasive melanomas (pT1, >pT1) included. MicroM were analyzed among only five to six dermoscopic patterns in four studies (De Giorgi et al. 2012; Seidenari et al. 2014; Dika et al. 2017; Megaris et al. 2018 )[12,14,17,19]. Five studies were conducted without the comparator group [8,9,12,16,19]. Studies conducted by Carli et al. 2003, Friedman et al. 2008, Abbasi et al. 2008, Pupelli et al. 2013, Drugge et al.2018 and Campos-do-Carmo et al. 2021 have chosen the melanocytic lesions as the comparator group [7,10,11,13,18,20]. Two of them have confronted the microM with melanocytic nevi of the same diameter [13,20]. Remaining studies have chosen differ-ent types of melanomas as the comparator group [3,6,14,15,17]. The differences concerned the ratio between in situ and invasive melanomas (Pizzichetta et al.; 2001; Helsing et al.2004; Seidenari et al.2014; Emiroglu et al.; 2014) or clinical presentation – difficult-to-diagnose melanoma versus clin-ically evident melanoma (Dika et al. 2017) [3,6,14,15,17].
The diagnostic feasibility of dermoscopic algorithms in small diameter melanomas were performed by Bono et al. 2006 (Menzies’ method sensitivity 83%, specificity 69%), Seidenari et al.2014 (7-point checklist and ABCD rule of Stoltz, data of sensitivity or specificity were not provided) Dika et al. 2017 (diagnostic sensitivity of pattern analysis 64.05%, 7-point checklist 61.08%, Menzies’ method 57.30%, ABCD rule of Stoltz 42.70%.) and Campos-do-Carmo et al. 2021 (modified ABC-point list algorithm sensitivity 61.8%, specificity 48.3%) [9,14,17,20]. Six studies (Bono et al. 1999, Carli et al. 2003, Bono et al. 2004 and 2006, Abbasi et al. 2008, and Campos-do-Carmo et al. 2021) compared the dermoscopic examination with the naked eye examination based on clinical ABCDE rule with wide range of sensitivity 36.6-83% and specificity 38-91%) [5,7-9,11,20].
In the light of the presented published studies, our research was conducted on uniform group of melanomas (in context of histopathological invasiveness), where the diameter was the only cut-off criterion for investigation. The methodology of our study was very detailed, as we evaluated ac-cording to all currently described patterns applied for melanocytic lesions. The reason for this deci-sion was influenced by a suspicion that microM might simulate the benign lesions [17,20-27]. Be-cause multiple tests were performed, the p-values of results were corrected using the Bonferroni method. We also conducted the diagnostic feasibility of the previously investigated and commonly known algorithm - the 7-point checklist, and a new one – the TADA algorithm, which has never been assessed before in microM [27-29].
Our study aimed to verify two hypotheses. The first hypothesis concerned the possibility of dermo-scopic differentiation of cutaneous melanomas proved histopathologically as in situ (pTis) and thin melanomas (pT1a) in terms of their diameter. The second hypothesis was aimed at the diagnostic feasibility of selected dermoscopic algorithms (7-point checklist and TADA algorithm) in the pur-pose to detect ≤5.0 mm sized small melanomas in histopathological category as pTis and pT1a.
- We can propose addition of the supplementary table with the review of the literature as needed (marked in the Introduction). We haven’t done it so far due to its large size. We kindly leave the decision to you if it will be valuable to include such table in this manuscript.

(5) Is the research design appropriate? Can be improved
- The research design was improved by precise description of the objectives and statistical analysis.
- Our study aimed to verify two hypotheses. The first hypothesis concerned the possibility of der-moscopic differentiation of cutaneous melanomas proved histopathologically as in situ (pTis) and thin melanomas (pT1a) in terms of their diameter. The second hypothesis was aimed at the diag-nostic feasibility of selected dermoscopic algorithms (7-point checklist and TADA algorithm) in the purpose to detect ≤5.0 mm sized small melanomas in histopathological category as pTis and pT1a.
(6) Are the methods adequately described? Can be improved
- The methodology of the study was revised and corrected. New description of the statistical anal-ysis and the study limitations were added.
(7) Are the conclusions supported by the results? Not applicable
- The conclusions were corrected to support the obtained results.
Conclusion: The dermoscopy of micromelanomas histopathologically staged as pTis and pT1a did not reveal the melanoma-specific pattern in 40% of them. Among the general melanocytic patterns, the spitzoid one was the frequently found in melanomas sized ≤5.0 mm. The 7-point checklist and TADA dermoscopic algorithms were helpful in the identification of the majority of sized ≤5.0 mm melanomas.

Reviewer 2 Report

This paper did a retrospective multicenter study on assessing dermoscopic features of histopathologically confirmed cutaneous melanomas in situ (pTis) and thin pT1a according to selected diagnostic algorithms. The dermoscopic images were evaluated by three independent investigators and the TADA algorithm as well as the 7-point algorithm were chosen for diagnostic feasibility. The authors found that a spitzoid dermoscopic pattern within lesions not exceeding 5 mm in 55 diameter should be an indication for their surgical removal due to the possible diagnosis of melanoma stage pTis to pT1a. This paper is with elaborate descriptions of the findings and well documented. I only have some basic concerns:

(1) The contributions or the novelty of this paper are not clearly presented; the authors should clearly demonstrate them.

(2) Compared to existing studies, what advancements did this paper make? It would be great if comparable studies can be done.

(3) Were there any p-value correction (or adjustment) methods applied to the statistical analysis in this study? Since multiple statistical significance tests were performed, p-value correction methods should be used.

Author Response

This paper did a retrospective multicenter study on assessing dermoscopic features of histopathologically confirmed cutaneous melanomas in situ (pTis) and thin pT1a according to selected diagnostic algorithms. The dermoscopic images were evaluated by three independent investigators and the TADA algorithm as well as the 7-point algorithm were chosen for diagnostic feasibility. The authors found that a spitzoid dermoscopic pattern within lesions not exceeding 5 mm in 55 diameter should be an indication for their surgical removal due to the possible diagnosis of melanoma stage pTis to pT1a. This paper is with elaborate descriptions of the findings and well documented.

 I only have some basic concerns:

(1)       The contributions or the novelty of this paper are not clearly presented; the authors should clearly demonstrate them.

- We have extensively revised and corrected the introduction, results and conclusions in aim to emphasize the novelty of this study in comparison to previous ones and underline the importance of the obtained results.  Moreover, we thoroughly attached into the introduction  additional citations with short summaries of the studies to highlight the lack of data in previous studies in comparison to obtained results and novel analysis (including TADA algorithm) in the study conducted by us.

Introduction

In the light of the presented published studies, our research was conducted on uniform group of melanomas (in context of histopathological invasiveness), where the diameter was the only cut-off criterion for investigation. The methodology of our study was very detailed, as we evaluated according to all currently described patterns applied for melanocytic lesions. The reason for this decision was influenced by a suspicion that microM might simulate the benign lesions [17,20-27]. Because multiple tests were performed, the p-values of results were corrected using the Bonferroni method. We also conducted the diagnostic feasibility of the previously investigated and commonly known algorithm - the 7-point checklist, and a new one – the TADA algorithm, which has never been assessed before in microM [27-29].

Discussion

In view of the results obtained in the context of adjusted p-value, we have extensively revised the entire subsection on discussion of results, relating the data obtained to the literature data, taking into account all valuable and pertinent suggestions made by the esteemed reviewers. Below are excerpts containing the main corrections.

- The most remarkable conclusion of our study was that one of the spitzoid patterns might be the first dermoscopic symptom of small melanomas (microM) (48% vs 11.86%, p=0.0004, ap=0.0013) although it was not reported in this frequency previously [14,19]. Seidenari et al. in the group of 22 micromelanomas sized ≤ 4mm displayed six dermoscopic patterns: multicomponent 32%, bicomponent 27%, spitzoid 18%, reticular 14%, globular and island 5% [14]. Megaris et al. displayed different frequency of the global patterns in melanomas with a diameter up to 5 mm including reticular 57.7%, structureless 26.9%, mixed (globular and reticular) 11.5%, starburst (spitzoid) 3,8% and none for globular and multicomponent ones [19].

- Next important remark of our study was finding that 40% of the microM versus 6.78% of larger melanomas (p=0.00007, ap=0.0023) did not present melanoma — specific patterns despite their in situ or early invasive stage (pT1a). This result indirectly proofed the diagnostic difficulties of the micromelanoma, as a potential imitator of the melanocytic nevi, which was reported in respect to micromelanomas and to melanomas in situ  [20,21,36-39]. 

- Our distinctive conclusion was that microM are almost deprived of negative network (p=0.04), shiny white structures (p=0.0027) and regression features (p=0.00003). Among vascular structures microM contained only dotted (32%) or polymorphous (28%) vessels, although rarely than melanoma sized > 5.0 mm (66.1% p=0.017 and 49% p>0.05 respectively).

- The diagnostic feasibility of the revised 7-point checklist in our study revealed the sensitivity of 60% and 98.31% in the micromelanoma group and the above 5 mm sized melanomas respectively (p<0.000001, ap=0.000006) (Table 5). The additional criteria i.e. 7 rules to avoid missing melanoma were applied in 72% of under 5 mm sized melanomas group versus 22.03% of larger ones (p<0.000001, ap=0.000006) and helped in diagnose of another 14 cases. The melanomas which complied with one or both of those criteria reached 88% in the micromelanoma group, indicating that only six of the micromelanomas were diagnosed as false negative. The TADA algorithm has never been evaluated in the micromelanoma group before. The results showed the melanoma-specific pattern present in 64% of microM group and in 96.6% in the comparative group (p=0.00001, ap=0.00006) and the melanoma-specific features in 72% (p>0.05), what finally leads to 3 microM cases that remain undiagnosed.

- Our study explored the dermoscopic morphology of micromelanomas among > 5 mm sized melanomas stage pTis and pT1a, indicating that application of the 7-point checklist including the 7 rules not to miss melanoma incognito or the TADA algorithms might help in revealing about 90% of micromelanomas.

- This study confirmed those findings, as the median of age in micromelanomas was 39.5 years, compared to 51.5 years in the comparative group (p=0.018), with the prevalence of female gender equally represented in both groups by about 60% (p>0.05). It seems that further education companies about the primary and secondary prevention of melanoma should be directed toward the male population which less frequently attends the screening dermoscopic visits.

(2)       Compared to existing studies, what advancements did this paper make? It would be great if comparable studies can be done.

- The comparison to all 18 published studies was performed in the introduction. In the discussion the results were confronted mainly with Seidenari et al.2014; Emiroglu et al.; 2014 Dika et al. 2017, Megaris et al.2018 and Campos-do-Carmo et al. 2021 studies with regard to the dermoscopic structures and patterns.

- The comparison of the diagnostic feasibility between the published studies was presented in the introduction and the discussion

- Introduction

- The diagnostic feasibility of dermoscopic algorithms in small diameter melanomas were performed by Bono et al. 2006 (Menzies’ method sensitivity 83%, specificity 69%), Seidenari et al.2014 (7-point checklist and ABCD rule of Stoltz, data of sensitivity or specificity were not provided) Dika et al. 2017 (diagnostic sensitivity of pattern analysis 64.05%, 7-point checklist 61.08%, Menzies’ method 57.30%, ABCD rule of Stoltz 42.70%.) and Campos-do-Carmo et al. 2021 (modified ABC-point list algorithm sensitivity 61.8%, specificity 48.3%) [9,14,17,20]. Six studies (Bono et al. 1999, Carli et al. 2003, Bono et al. 2004 and 2006, Abbasi et al. 2008, and Campos-do-Carmo et al. 2021) compared the dermoscopic examination with the naked eye examination based on clinical ABCDE rule with wide range of sensitivity 36.6-83% and specificity 38-91%) [5,7-9,11,20].

Discussion:

- Dika et al. as the only ones performed the diagnostic feasibility of four diagnostic algorithms [17]. The sensitivity of modified pattern analysis, ABCD rule of dermoscopy, 7-point check list of Argenziano and Menzies’ method for thin melanomas of the lower limbs was lower than in the previous studies and reached 64.05%, 42.70%, 61.80% and 57.30% respectively [17]. Pupelli et al. analyzed the 7-point check list and obtained a positive diagnose (score =/>3 points) in 22/24 of melanomas sized under 5 mm in diameter [13]. The diagnostic feasibility of the revised 7-point checklist in our study revealed the sensitivity of 60% and 98.31% in the micromelanoma group and the above 5 mm sized melanomas respectively (p<0.000001, ap=0.000006) (Table 5). The additional criteria i.e. 7 rules to avoid missing melanoma were applied in 72% of under 5 mm sized melanomas group versus 22.03% of larger ones (p<0.000001, ap=0.000006) and helped in the diagnosis of another 14 cases. The melanomas which complied with one or both of those criteria reached 88% in the micromelanoma group, indicating that only six of the micromelanomas were diagnosed as false negative. The TADA algorithm has never been evaluated in the micromelanoma group before. The results showed the melanoma-specific pattern present in 64% of microM group and in 96.6% in the comparative group (p=0.00001, ap=0.00006) and the melanoma-specific features in 72% (p>0.05), what finally leads to 3 microM cases that remain undiagnosed.

(3)       Were there any p-value correction (or adjustment) methods applied to the statistical analysis in this study? Since multiple statistical significance tests were performed, p-value correction methods should be used.

– The statistical analysis was completely revised and corrected.

The statistical analysis was performed using the Statistica version 10.0 software for Windows. The Pearson's Chi2, Fisher exact two-tailed test  and the crosstabulation tables were applied for the frequency statistics.  The nonparametric Wald-Wolfowitz test runs and Mann-Whitney U test for independent samples were applied for numerical data. Odds ratios and corresponding 95% confidence limits (CL) were calculated by the logistic regression adjusted for age for the melanoma specific dermoscopic structures, the dermoscopic patterns and for the diagnostic feasibility of selected algorithms. Because multiple tests were performed, the p-values of results were corrected using Bonferroni method. The statistical significance of p < 0.05 was considered.

(4) Moderate English changes required

- The manuscript was revised by the English native speaking person. All corrections have been made in the main text in the change tracking mode.

(5) Does the introduction provide sufficient background and include all relevant references? Can be improved

- The introduction was completely revised and the background with summary of all 18 published studies added to the text. We have described the design and general results of those studies. In the discussion we have reviewed their results in detail.

In recent decades, dermoscopy has proved to be irreplaceable in the non-invasive diagnosis of melanoma, as well as established in skin malignant neoplasms’ secondary prevention [1]. Small-sized in diameter cutaneous melanomas, also called micromelanomas (microM), were reported with variable frequency within the range of 1.5-38.2%, depending on the threshold value obtained and the number of samples included in the study [2,3]. According to Akay et al., the smallest microM that has been detected by dermoscopic examination so far had a diameter of 0.9 x 0.6 mm [4].

The first publication describing the dermoscopic diagnosis of microM was carried out by Bono et al. 1999 [5]. So far, seventeen other studies have been published differing significantly in the size of the sample (8-123 microM; median 27.5) and methodology [2,3,6-20] (Supplementary Table).

Unfortunately the results regarding the diagnostic frequency of dermoscopic structures are difficult to compare, due to differences in the number of selected  dermoscopic structures, chosen comparator and different ratio of in situ to invasive melanomas (pT1, >pT1) included. MicroM were analyzed among only five to six dermoscopic patterns in four studies (De Giorgi et al. 2012; Seidenari et al. 2014; Dika et al. 2017; Megaris et al. 2018 )[12,14,17,19].  Five studies were conducted without the comparator group [8,9,12,16,19]. Studies conducted by Carli et al. 2003, Friedman et al. 2008, Abbasi et al. 2008, Pupelli et al. 2013, Drugge et al.2018 and Campos-do-Carmo et al. 2021 have chosen the melanocytic lesions as the comparator group [7,10,11,13,18,20]. Two of them have confronted the microM with melanocytic nevi of the same diameter [13,20]. Remaining studies have chosen different types of melanomas as the comparator group [3,6,14,15,17].  The differences concerned the ratio between in situ and invasive melanomas (Pizzichetta et al.; 2001; Helsing et al.2004; Seidenari et al.2014; Emiroglu et al.; 2014) or clinical presentation – difficult-to-diagnose melanoma versus clinically evident melanoma (Dika et al. 2017) [3,6,14,15,17].

The diagnostic feasibility of dermoscopic algorithms in small diameter melanomas were performed by Bono et al. 2006 (Menzies’ method sensitivity 83%, specificity 69%), Seidenari et al.2014 (7-point checklist and ABCD rule of Stoltz, data of sensitivity or specificity were not provided) Dika et al. 2017 (diagnostic sensitivity of pattern analysis 64.05%, 7-point checklist 61.08%, Menzies’ method 57.30%, ABCD rule of Stoltz 42.70%.) and Campos-do-Carmo et al. 2021 (modified ABC-point list algorithm sensitivity 61.8%, specificity 48.3%) [9,14,17,20]. Six studies (Bono et al. 1999, Carli et al. 2003, Bono et al. 2004 and 2006, Abbasi et al. 2008, and Campos-do-Carmo et al. 2021) compared the dermoscopic examination with the naked eye examination based on clinical ABCDE rule with wide range of sensitivity 36.6-83% and specificity 38-91%) [5,7-9,11,20].

In the light of the presented published studies, our research was conducted on uniform group of melanomas (in context of histopathological invasiveness), where the diameter was the only cut-off criterion for investigation. The methodology of our study was very detailed, as we evaluated according to all currently described patterns applied for melanocytic lesions. The reason for this decision was influenced by a suspicion that microM might simulate the benign lesions [17,20-27]. Because multiple tests were performed, the p-values of results were corrected using the Bonferroni method. We also conducted the diagnostic feasibility of the previously investigated and commonly known algorithm - the 7-point checklist, and a new one – the TADA algorithm, which has never been assessed before in microM [27-29].

Our study aimed to verify two hypotheses. The first hypothesis concerned the possibility of dermoscopic differentiation of cutaneous melanomas proved histopathologically as in situ (pTis) and thin melanomas (pT1a) in terms of their diameter. The second hypothesis was aimed at the diagnostic feasibility of selected dermoscopic algorithms (7-point checklist and TADA algorithm) in the purpose to detect ≤5.0 mm sized small melanomas in histopathological category as pTis and pT1a.

(6) Is the research design appropriate? Can be improved

- The research design was improved by precise description of the objectives and statistical analysis. Limitations of the study were also corrected.

- Our study aimed to verify two hypotheses. The first hypothesis concerned the possibility of dermoscopic differentiation of cutaneous melanomas proved histopathologically as in situ (pTis) and thin melanomas (pT1a) in terms of their diameter. The second hypothesis was aimed at the diagnostic feasibility of selected dermoscopic algorithms (7-point checklist and TADA algorithm) in the purpose to detect ≤5.0 mm sized small melanomas in histopathological category as pTis and pT1a.

- Limitations of the study: The first limitation of our study is the retrospective analysis and lack of the comparison with the benign melanocytic lesions including small diameter nevi. The bias of the study was the selective inclusion of melanomas with fully archived videodermoscopic records. Other bias of the study was the profile of patients from dermatologic centres, where the small diameter melanomas predominate, as opposed to oncology centres, where patients usually present at more advanced clinical and histopathologic stages. The material came from exclusively dermatological centres where the same methods of demoscopic examination and archiving were used therefore no differences were found in the analyzed patient population.

(7) Are the methods adequately described? Can be improved

- The methodology of the study was revised and corrected. New description of the statistical analysis and the study limitations were added.

(8) Are the results clearly presented? Can be improved

- The results were revised according to the implemented statistical analysis (Fisher exact test, Bonferroni correction, logistic regression). New data were added. All obtained results according to adjusted p-value was added in  all tables. Therefore all tables were revised according to results.

(9) Are the conclusions supported by the results? Can be improved

The conclusions were corrected to support the obtained results adequately.

Abstract

Conclusion: The dermoscopy of micromelanomas histopathologically staged as pTis and pT1a did not reveal the melanoma-specific pattern in 40% of them. Among the general melanocytic patterns, the spitzoid one was the frequently found in melanomas sized ≤5.0 mm. The 7-point checklist and TADA dermoscopic algorithms were helpful in the identification of the majority of sized ≤5.0 mm melanomas.

Body of manuscript

Conclusions

The dermoscopy of micromelanomas staged as pTis and pT1a did not reveal the melanoma-specific pattern in 40% of the microM (p=0.00007, ap=0.0023). Among the general melanocytic patterns, the spitzoid one was the most frequently found in microM (48% vs 11.86%, p =0.0004, ap=0.0013). Knowledge of these dermoscopic features brings the clinician closer to an early diagnosis of melanoma with a diameter of 5 mm or less. The 7-point checklist and TADA dermoscopic algorithms were helpful in diagnosis of the majority.

Round 2

Reviewer 1 Report

Dear authors, thank you for the thorough revision of your paper.

Author Response

Most Esteemed Reviewer, 
On behalf of myself and the co-authors, we would like to thank you very much for kindly accepting our corrections. The comments were extremely valuable and certainly informative for us. We send you our warmest greetings and express our gratitude